Size matters: relationships between body size and body mass of common coastal, aquatic invertebrates in the Baltic Sea

Eklöf Johan johan.eklof@su.se 1
Austin Åsa 1
Bergström Ulf 2
Donadi Serena 1 3
Eriksson Britas D.H.K. 4
Hansen Joakim 3
Sundblad Göran 5
1 Department of Ecology, Environment and Plant Sciences (DEEP), Stockholm University , Stockholm , Sweden
2 Department of Aquatic Resources, Swedish University of Agricultural Sciences , Öregrund , Sweden
3 Baltic Sea Centre, Stockholm University , Stockholm , Sweden
4 Groningen Institute for Evolutionary Life-Sciences GELIFES, University of Groningen , Groningen , Netherlands
5 Aquabiota Water Research , Stockholm , Sweden
Taylor Richard
Electronic publication date: 2017 Jan 25
Publication date: 2017
Volume: 5
Electronic Location ID: e2906
Received 2016 Aug 25; Accepted 2016 Dec 12
Copyright: ©2017 Eklöf et al.
Copyright year: 2017
Copyright holder: Eklöf et al.
License: This is an open access article distributed under the terms of the Creative Commons Attribution License, which permits unrestricted use, distribution, reproduction and adaptation in any medium and for any purpose provided that it is properly attributed. For attribution, the original author(s), title, publication source (PeerJ) and either DOI or URL of the article must be cited.
License URL: https://creativecommons.org/licenses/by/4.0/

Keywords: Estuary, Biometry, Infauna, Submerged aquatic vegetation, Isometric scaling, Length:weight relationship, Epifauna, Allometry, Seagrass, Weight

Funding: Formas 2013-1074 His Majesty Carl XVI Gustaf’s Foundation for Science and Education HMK-2014-0002 Baltic Sea 2020 foundation Stockholm University Baltic Sea Centre Groningen University This study was funded by Formas (grant 2013-1074), His Majesty Carl XVI Gustaf’s Foundation for Science and Education (grant HMK-2014-0002), the Baltic Sea 2020 foundation (project “Levande kust”), Stockholm University Baltic Sea Centre (Askö grants and support to MSc students), and in-kind support from Groningen University (Netherlands) to BDHK Eriksson. The funders had no role in study design, data collection and analysis, decision to publish, or preparation of the manuscript.

==============================
Background

Organism biomass is one of the most important variables in ecological studies, making biomass estimations one of the most common laboratory tasks. Biomass of small macroinvertebrates is usually estimated as dry mass or ash-free dry mass (hereafter ‘DM’ vs. ‘AFDM’) per sample; a laborious and time consuming process, that often can be speeded up using easily measured and reliable proxy variables like body size or wet (fresh) mass. Another common way of estimating AFDM (one of the most accurate but also time-consuming estimates of biologically active tissue mass) is the use of AFDM/DM ratios as conversion factors. So far, however, these ratios typically ignore the possibility that the relative mass of biologically active vs. non-active support tissue (e.g., protective exoskeleton or shell)—and therefore, also AFDM/DM ratios—may change with body size, as previously shown for taxa like spiders, vertebrates and trees.

Methods

We collected aquatic, epibenthic macroinvertebrates (>1 mm) in 32 shallow bays along a 360 km stretch of the Swedish coast along the Baltic Sea; one of the largest brackish water bodies on Earth. We then estimated statistical relationships between the body size (length or height in mm), body dry mass and ash-free dry mass for 14 of the most common taxa; five gastropods, three bivalves, three crustaceans and three insect larvae. Finally, we statistically estimated the potential influence of body size on the AFDM/DM ratio per taxon.

Results

For most taxa, non-linear regression models describing the power relationship between body size and (i) DM and (ii) AFDM fit the data well (as indicated by low SE and high R2). Moreover, for more than half of the taxa studied (including the vast majority of the shelled molluscs), body size had a negative influence on organism AFDM/DM ratios.

Discussion

The good fit of the modelled power relationships suggests that the constants reported here can be used to quickly estimate organism dry- and ash-free dry mass based on body size, thereby freeing up considerable work resources. However, the considerable differences in constants between taxa emphasize the need for taxon-specific relationships, and the potential dangers associated with ignoring body size. The negative influence of body size on the AFDM/DM ratio found in a majority of the molluscs could be caused by increasingly thicker shells with organism age, and/or spawning-induced loss of biologically active tissue in adults. Consequently, future studies utilizing AFDM/DM (and presumably also AFDM/wet mass) ratios should carefully assess the potential influence of body size to ensure more reliable estimates of organism body mass.

Introduction

Organism biomass is inarguably one of the more important variables in ecology, playing a central role in studies ranging from ecophysiology and community and food web regulation, to whole-ecosystem metabolism (e.g.,  Enquist & Niklas, 2001; Gruner et al., 2008; Perez-Harguindeguy et al., 2013). As a consequence, to accurately estimate organism biomass constitutes one of the most common and important tasks in ecological studies.

Small invertebrates retained on 0.5–1 mm sieves (hereafter ‘macrofauna’) make up a major part of animal density, diversity and biomass in many ecosystems; e.g., insects and arachnids in terrestrial ecosystems; epibenthic, aquatic crustaceans, echinoderms and molluscs in stands of aquatic vegetation; and infaunal (sediment-dwelling) worms, crustaceans and molluscs in marine sediments. Macrofauna biomass is typically reported as dry- or ash-free dry mass per unit area (e.g.,  g per m2), which requires observers to repeatedly identify, sort, dry and weigh individual or pooled organisms; a time-consuming, expensive and tedious process. Many studies have shown that more easily measured proxy variables scale predictably with dry mass and therefore can be used to speed up biomass estimations; e.g.,  wet (fresh) mass (Brey, Rumohr & Ankar, 1988; Ricciardi & Bourget, 1998; Brey et al., 2010) and body size, based on either exact length measurement (Smock, 1980; Frithsen, Rudnick & Doering, 1986; Sabo, Bastow & Power, 2002) or retention on sieves of certain mesh sizes (Widbom, 1984; Edgar, 1990; Casagranda & Boudouresque, 2002). While wet mass can be a very good proxy (see e.g.,  Ricciardi & Bourget, 1998), we—as others before—argue that body size (e.g.,  length) holds several advantages, and lacks several disadvantages associated with wet mass estimations. First, ecological theory supported by empirical data suggest body mass scales predictably with length in the form of power relations (Smock, 1980; Sabo, Bastow & Power, 2002). Second, while freezing/thawing and fixation in conservation liquids (e.g.,  EtOH or formalin) can affect both organism wet mass (Howmiller, 1972; Mason, Lewis & Weber, 1983; Leuven, Brock & van Druten, 1985) and length (Hjörleifsson & Klein-MacPhee, 1992; Kapiris, Miliou & Moraitou-Apostolopoulou, 1997), wet mass estimations are also very sensitive to exactly how specimens are blotted, centrifuged (to remove excess water), and exposed to air and light before and during weighing (Howmiller, 1972; Mason, Lewis & Weber, 1983; Leuven, Brock & van Druten, 1985). As size estimations do not require blotting, they are less sensitive to observer error, and also faster to perform. Third, body size (e.g.,  length or height) estimations can more easily be automated, using e.g.,  image analysis software (Paavo et al., 2008; Mallard, Le Bourlot & Tully, 2013), to rapidly process multiple individuals at a time.

In benthic ecology, ash-free dry mass (hereafter ‘AFDM’, in the older literature sometimes called ‘ash-free dry weight’ or simply ‘AFDW’) is often regarded as the most accurate predictor of macrofauna biomass, as it only includes biologically active tissue. Since AFDM estimations require the incineration of dried samples in a furnace at high temperature, adding considerable time and costs to analyses, many studies have reported how AFDM scales with estimations of wet- and dry mass, usually in the form of simple ratios as ‘conversion factors’ (e.g., AFDM/DM, in %) (Rumohr, Brey & Ankar, 1987; Ricciardi & Bourget, 1998). However, these ratios typically ignore the possibility that the relative mass of biologically active vs. non-active support tissue (e.g.,  protective exoskeleton or shell)—and therefore the AFDM/DM ratio—may change with macrofauna body size, as previously shown for disparate taxa like spiders (Andersen, 1979), vertebrates (Miller & Birchard, 2005) and trees (Niklas, 1995). This issue is important not only for obtaining accurate biomass conversions and estimations, but also for understanding how organismal investment in one type of structure may limit or constrain investment in other structures across ontogenetic development stages (Lease & Wolf, 2010).

Here we estimate and report relationships between body size, dry mass and ash-free dry mass for 14 of the most common aquatic, epibenthic invertebrate taxa found in shallow, vegetated habitats of the central Baltic Sea; one of the largest brackish water bodies on Earth. For each taxon we also assess whether the ash-free dry mass/dry mass ratio changes with body size. Our aim is to provide simple yet reliable size-based relationships that can be used to rapidly estimate organism body mass and, ultimately, biomass per sample.

Methods

Study area

The Baltic Sea is a 415,000 km2 large marginal sea situated in northern Europe (53–66°N; 10–30°E). A main feature is the presence of strong horizontal and vertical gradients in salinity, temperature and oxygen, that also undergo considerable temporal (e.g.,  seasonal) fluctuations (Voipio, 1981). The Baltic Sea is evolutionary very young (ca 6,000 years), and the shallow coastal areas have since the last glaciation been colonized by a mixture of marine, freshwater and brackish organisms, including crustaceans, gastropods, bivalves, polychaetes, hirudineans, nemerteans and insect larvae (Hansen, Wikström & Kautsky, 2008). As many marine and freshwater organisms in the Baltic Sea live near their physiological tolerance limits, they grow slower and smaller than in their original environment; e.g.,  the blue mussel Mytilus edulis (Tedengren & Kautsky, 1986). As a consequence, their size ranges—but also size:mass relationships and, potentially, AFDM/DM ratios—could differ from those reported for conspecifics in marine or freshwater areas (Rumohr, Brey & Ankar, 1987). An estimate of the effect of salinity on size:mass or DM:AFDM relationship was beyond the scope of our study, but our results could be compared to relationships in marine populations of the same taxa, if sampled and measured in the same way.

Field sampling

During summer (May–Aug) 2014 we collected aquatic invertebrate macrofauna (>1 mm) in 32 shallow bays situated along a 360 km stretch of the central, Swedish Baltic Sea coastline (Fig. 1). The salinity in the area is generally low (ca. 5–7 psu) but fluctuates strongly with freshwater runoff and upwelling events. In each bay, a snorkeler sampled submerged aquatic vegetation and epibenthic macrofauna in 3–8 randomly selected stations (>30 m apart), by gently placing a 20 × 20 cm frame (with a 1 mm-mesh bag attached) on the sea bed, and collecting all organisms (aquatic vegetation and associated invertebrates) found above or on top of the sediment surface. The bag content was immediately transferred to a plastic bag, which was kept cold on ice until frozen (−20 °C), in most cases within 1–3 h.

Figure 1 Maps of Scandinavia (small image) and the sampling area.

Filled circles mark the position of the 32 sampled bays. Numbers along the x- and y-axis are longitude and latitude, respectively.

Body size estimations

Following thawing in room temperature, we identified intact invertebrate organisms to the highest taxonomic resolution feasible using standard literature. For the 14 most common taxa we then selected and measured the body size of 12–459 individuals per taxa (3,220 individuals in total), chosen to capture the full range of body sizes found across the 32 bays. The taxa included five gastropods (Theodoxus fluviatilis, Hydrobia spp., Radix balthica, Potamopyrgus antipodarum, Bithynia tentaculata), three bivalves (Mytilus edulis, Limecola (Macoma)) balthica and Cardidae spp. (numerically dominated by Parvicardium hauniense), three crustaceans (Amphibalanus improvisus, Idotea spp., Gammarus spp.) and three insects (larval stages of Chironomidae spp., Agraylea spp. and Phryganeidae spp.) (see also Table 1). Body size (to the nearest 1 mm) was measured (based on standard procedures; Hayward & Ryland, 1995) as; (i) gastropod height along the central shell axis, (ii) bivalve length from anterior to posterior side, (iii) total length of Gammarus and Idotea spp. from tip of rostrum to last urosome, (iv) body width for Amphibalanus improvisus, and (v) total length of insect larvae from end of head to last segment. A higher size accuracy is definitely possible (e.g., to 0.1 or 0.01 mm using calipers or stereo lenses), but as most studies utilizing this type of data (including ours) will depend on 1000s of length measurements, the accuracy chosen was a realistic trade-off between time and precision.

Table 1 Results of regression analyses estimating (i) the non-linear power relationship between body size and dry mass (DM) and (ii) ash-free dry mass (AFDM), (iii) the mean ± 1SE AFDM/DM ratio (in %), and (iv) the linear relationship between body size and AFDM/DM ratio (in %), for 14 macroinvertebrate taxa in shallow coastal areas of the Baltic Sea.

Letters within parentheses after taxa names denote classes.

		Body size vs. DM	Body size vs. AFDM	AFDM/DM	Body size vs. AFDM/DM	
Taxon	N	α ± SE	β ± SE	R2	α ± SE	β ± SE	R2	Mean % ± 1 SE	Intercept ± SE	Slope ± SE	R2	
Bithynia tentaculata L. (G)	25	0.598 ± 0.484ns	2.117 ± 0.351***	0.847	0.479 ± 0.511ns	1.36 ± 0.472**	0.668	19.133 ± 2.207	33.162 ± 3.878***	−1.91 ± 0.452*	0.424	
Hydrobia spp. (G)	24	0.239 ± 0.041***	2.134 ± 0.095***	0.952	0.079 ± 0.029*	1.441 ± 0.22***	0.758	13.737 ± 1.155	19.791 ± 2.855***	−0.633 ± 0.715*	0.155	
Potamopyrgusantipodarum Gray (G)	17	0.479 ± 0.511ns	1.360 ± 0.472**	0.919	0.021 ± 0.012ns	2.447 ± 0.395***	0.898	16.051 ± 1.399	6.063 ± 4.616ns	2.653 ± 1.180*	0.202	
Radix balthica L. (G)	20	0.137 ± 0.035**	2.355 ± 0.115***	0.956	0.046 ± 0.018*	2.119 ± 0.177***	0.906	27.087 ± 2.233	35.338 ± 3.558***	−1.794 ± 0.650*	0.258	
Theodoxus fluviatilis L. (G)	29	0.221 ± 0.065**	2.683 ± 0.148***	0.9492	0.015 ± 0.006*	2.915 ± 0.194***	0.912	13.044 ± 1.083	18.52 ± 2.396***	−0.242 ± 0.494*	0.159	
Cardidae spp. (B)	33	0.134 ± 0.094ns	2.848 ± 0.347***	0.924	0.014 ± 0.013ns	2.806 ± 0.486***	0.879	12.358 ± 0.852	18.075 ± 1.468***	−0.429 ± 0.325*	0.364	
Limecola balthica L. (B)	18	0.069 ± 0.024*	2.820 ± 0.134***	0.991	0.001 ± 0.002ns	3.479 ± 0.673***	0.92	12.717 ± 1.934	21.429 ± 2.98***	−0.264 ± 0.372*	0.383	
Mytilus edulis L. (B)	24	0.030 ± 0.015*	2.933 ± 0.153***	0.991	0.006 ± 0.003*	2.844 ± 0.147***	0.978	14.189 ± 0.504	13.162 ± 1.044***	0.078 ± 0.069ns	0.011	
Amphibalanusimprovisus Darwin (C)	13	0.314 ± 0.205ns	2.515 ± 0.289***	0.976	0.036 ± 0.022ns	2.289 ± 0.276***	0.961	8.939 ± 0.550	11.044 ± 1.064***	−0.397 ± 0.179*	0.246	
Gammarus spp. (C)	37	0.047 ± 0.032ns	2.111 ± 0.265***	0.926	0.033 ± 0.028ns	2.05 ± 0.32***	0.863	58.966 ± 1.519	63.062 ± 2.616***	−0.389 ± 0.307ns	0.017	
Idothea spp. (C)	42	0.001 ± 0.001ns	3.592 ± 0.200***	0.949	0.001 ± 0.001ns	3.850 ± 0.249***	0.919	61.505 ± 1.659	66.183 ± 3.457***	−0.550 ± 0.358ns	0.032	
Agraylea spp. (I)	13	0.001 ± 0.002ns	3.410 ± 0.721**	0.820	0.001 ± 0.002ns	3.432 ± 0.769***	0.833	85.967 ± 3.769	88.893 ± 7.725***	0.570 ± 1.277ns	−0.097	
Chironomidae spp. (I)	38	0.014 ± 0.016ns	1.383 ± 0.290***	0.600	0.008 ± 0.006ns	1.544 ± 0.321***	0.533	79.307 ± 2.643	78.633 ± 6.947***	0.070 ± 0.688ns	−0.027	
Phryganeidae spp. (I)	10	0.001 ± 0.001ns	3.176 ± 0.649***	0.746	0.001 ± 0.001ns	3.207 ± 0.611***	0.789	91.851 ± 2.137	86.64 ± 3.558***	0.382 ± 0.185ns	0.290	
Notes.

G Gastropoda

B Bivalvia

C Crustacea

I Insecta (larvae)

α and β normalization and scaling constant for power equations, respectively

ns p < 0.05.

* p < 0.05.

** p < 0.01.

** p < 0.001

Values in bold mark those significant (at α = 0.05). Note: R2 were derived from linear log–log models.

Estimations of dry- and ash-free dry mass

Following size estimations, the measured individuals were transferred to pre-dried and -weighed (nearest 0.0001 g) porcelain crucibles. For most size classes (except for very large and rare individuals), multiple individuals were typically pooled into the same crucible. This step underestimates actual variability in body mass between individuals, but was necessary as the low individual body masses (particularly AFDM) were near or below the reliable detection limit of the scale. We included multiple estimations of the same sizes, so that the number of biomass estimations (N) ranged from 10 to 42 per taxa. Samples were then dried at 60 °C for >48 h (until constant mass), and cooled to room temperature in a desiccator before weighing. To estimate ash-free dry mass, the crucibles were then transferred to a muffle furnace, incinerated (550 °C for 3 h), cooled and weighed again. Ash-free dry mass was calculated as dry mass minus ash mass.

Statistical analyses

We estimated taxon-specific body size:body mass relationship using non-linear regression in the form of the power equation: body mass=α×sizeβ

where body mass is the individual mass (mg DM or AFDM), size is the body size (length/height, in mm), α is a normalization constant, and β is the scaling constant. Body mass typically scales with size in a power relationship, and initial data exploration showed that power equations provided a superior fit compared to linear, log or exponential relationships. As regression coefficients (R2) are an inadequate measure of fit for non-linear regression models (Spiess & Neumeyer, 2010), we report SE for α and β. However, for the sake of simplicity we also estimated the linear log–log relationship between body size and biomass, and report the R2 for those models (see e.g., Lease & Wolf, 2010).

For each taxon we also calculated the mean (±1 SE) AFDM/DM ratio (in %); a commonly used conversion factor in macroinvertebrate studies (see e.g., Ricciardi & Bourget, 1998). We then used linear regression to test whether body size (in mm) affected the AFDM/DM ratio. Prior to analyses we checked assumptions of normality (by plotting predicted vs. observed quantiles) and homoscedasticity (by plotting predicted vs. observed residuals). All analyses were conducted in R v. 3.2.3 (R Core Team, 2016).

Results

Relationships between body size and individual biomass

The relationships between body size (mm), individual dry mass (mg DM) and ash-free dry mass (mg AFDM) for all 14 taxa are displayed in Figs. 2A–2H, and the parameters (and their fit) are presented in Table 1. For most of the taxa, body size was a very good predictor of individual DM, as demonstrated by low SE and R2 near 1. The model fits were slightly poorer for the three insect taxa (R2 = 0.60–0.82) and the gastropod Bithynia tentaculata (R2 = 0.85) than for the other ten taxa. For a majority (12 out of 14) of the taxa, the scaling constants (β) were well above 2 (2.110–3.590). The exceptions were the small gastropod Potamopyrgus antipodarum and chironomid larvae, which had constants closer to 1 (β = 1.368 and 1.383, respectively).

Figure 2 Best-fitting relationships between body size (length or height, see ‘Methods’) and (A–D) dry mass (mg. DM), (E–H) ash-free dry mass (mg. AFDM) and (I–L) AFDM/DM ratio (% AFDM), for 14 taxa—five gastropods, three bivalves, three crustaceans and three insect larvae—sampled in coastal areas of the central Baltic Sea.

For model parameters and estimates of fit, see Table 1.

Body size was also a very good predictor of AFDM, even though model fits (based on SE and R2) were slightly poorer than for DM (Table 1). Just as for DM relationships, the model fits (based on SE and R2) were best for gastropods, molluscs and crustaceans. The scaling constants (β) were for most taxa quite similar to those reported for the DM relationships, with the exception of a higher constant for P. antipodarum (β = 2.447) and a lower constant for Bithynia tentaculata (β = 1.360).

Influence of organism body size on AFDM/DM ratios

The AFDM/DM ratios (mean % ± SE) per taxa are also presented in Table 1. As expected, there were consistent differences between the four major taxonomic groups studied, with low AFDM content in bivalves and gastropods (12–27%), who’s calcium carbonate shell makes up the major part of whole-body biomass, to higher AFDM content in chitin-shelled crustaceans (ca 60%), and the highest content in insect larvae (86–92%).

Results of simple linear regression showed that for more than half (8 out of 14) of the taxa surveyed, body size clearly affected the AFDM/DM ratio (Table 1 and Figs. 2I–2L). For four out of five gastropods, two out of three bivalves, as well as the sessile, calcite-shelled crustacean Amphibalanus improvisus, the AFDM/DM ratio decreased linearly with body size. For the small gastropod Potamopyrgus antipodarum body size instead had a positive influence on AFDM/DM. However, the P. antipodarum size range was very narrow (2–4 mm) and the intercept was not different from 0 (Table 1), suggesting a relatively poor model fit. Moreover, there was no size effect found in the blue mussel Mytilus edulis (Table 1). Finally, in contrast to the size effects found for most of the hard-shelled molluscs, there was no influence of body size on AFDM/DM in any of the chitin-shelled crustaceans or insect larvae (Table 1 and Figs. 2I–2L).

Discussion

Estimating organism biomass is one of the most common, important but also resource-consuming tasks in ecological work, particularly when it comes to small-bodied, highly abundant and diverse macroscopic invertebrates. Many previous studies have shown that more easily measured variables like invertebrate wet (fresh) mass (e.g., Ricciardi & Bourget, 1998) or body size (e.g., Smock, 1980) can be used as proxies to reliably predict both the dry- and ash-free dry body mass, thereby simplifying and speeding up biomass estimations. Here, we first complement this literature by reporting how body mass scales with body size for 14 of the most common epibenthic invertebrate taxa found in shallow coastal areas of the Baltic Sea. Moreover, we demonstrate that for a majority of the studied molluscs, the ratio between organism dry- and ash-free dry mass—an often-used conversion factor (e.g., Rumohr, Brey & Ankar, 1987; Ricciardi & Bourget, 1998)—decreases predictably with body size. Thus, our results can be used to quickly estimate the biologically active biomass of individual organisms based on their size, and when combined with density data, accurately estimate biomass per unit area.

Body size as a proxy for dry- and ash-free dry mass

For a majority of the studied taxa, body size was a good predictor of both dry mass and ash-free dry mass. The model fits were slightly poorer for ash-free dry mass (AFDM); most likely a consequence of the fact that even though multiple individuals of the same size were pooled, the low individual AFDM of many organisms (in the vicinity of 1 mg) challenged the accuracy of the scale. Comparisons between the 14 taxa studied (Table 1) show that particularly within the gastropods and crustaceans, the scaling (β) constants differ quite substantially between taxa (see the different slopes in Fig. 2 and β coefficients in Table 1). These differences emphasize the need for taxon-specific relationships to accurately predict biomass, and the potential dangers in either ignoring body size or substituting relationships between taxa. Consequently, our power equations (Table 1) can be used in a simple yet reliable way to estimate organism dry- or ash-free dry mass based on standard body size measurements. Future studies should ideally also assess how these relationships vary in time and space (e.g., over seasons), for even more accurate biomass estimations. Size-based biomass estimations are likely to speed up laboratory work considerably; for example, Casagranda & Boudouresque (2002) showed that sieve-based size estimations speeded up estimations of body biomass of the gastropod Hydrobia ventricosa by 20–30 times. Consequently, our size-based estimations of invertebrate biomass are likely to free up considerable work resources (time, man-power, money) that can be used to e.g., collect and process more samples.

The influence of body size on AFDM/DM ratios

For most of the taxa with a calcium-carbonate (molluscs) or calcite shell (the barnacle Amphibalanus improvisus), we found a significant negative influence of body size on the AFDM/DM ratio; a commonly reported and often-used conversion factor in macrofauna studies (e.g., Rumohr, Brey & Ankar, 1987; Ricciardi & Bourget, 1998). In other words, the proportional mass of biologically active vs. non-active tissue (shell, hard mouth parts, etc.) decreased with body size. There are at least two possible and complementary explanations for this relationship. First, while the rate of growth in length of mollusc shells typically decreases with age, new shell layers are consistently added on a yearly basis (Negus, 1966). This results in increasingly thicker, and therefore disproportionally heavier, shells with mussel length, and a higher shell:tissue mass ratio. Second, our sampling was conducted during summer; a season when a majority of adult molluscs (here represented by the larger individuals per taxa) most likely had spawned and temporarily lost a considerable proportion of their biologically active tissue (Kautsky, 1982). The slopes of the significant regressions (Table 1, median = − 1.26) suggest that failing to incorporate the potential influence of body size can strongly reduce the accuracy of AFDM estimations based on dry mass (and presumably also wet mass), particularly if there is considerable variability in body size in the samples. The somewhat surprising lack of size influence in the common blue mussel Mytilus edulis was not investigated in detail, but could be caused by (i) the lack of small shell-crushing mussel predators in the area (e.g., crabs), who otherwise are known to trigger thicker mussel shells (Freeman, 2007), and/or (ii) the relatively low salinity, which causes the small, osmotically stressed M. edulis to invest considerably more energy into osmosis and soft tissue production, than in thicker shells (Kautsky, Johannesson & Tedengren, 1990).

In contrast to the results for molluscs, there was no size effect on AFDM/DM ratios for the chitin-shelled insects and crustaceans. These results fit well with those reported in previous studies, for example of terrestrial insects, for which exoskeletal chitin scales isometrically (1:1) with body size (Lease & Wolf, 2010). In summary, our results suggest that body size can play an important but hitherto underestimated role when estimating organism AFDM based on dry (and possibly, wet) mass, particularly for shelled molluscs.

Conclusions

Using samples of epibenthic macroinvertebrates collected in 32 shallow bays along a 360 km stretch of the Swedish Baltic Sea coast, we show that for 14 of the most common macrofauna taxa, organism body size scales predictably with individual dry mass and ash-free dry mass in the form of power relations. The good model fits suggest the taxon-specific equations reported here can be used to predict individual biomass based on organism size, thereby speeding up estimations of macrofauna biomass. Moreover, for the vast majority of the molluscs studied, we find a negative relationship between body size and AFDM/DM ratio; a commonly used conversion factor in macrofauna studies. Consequently, future studies utilizing AFDM/DM ratios should carefully assess the potential influence of body size and spatial–temporal variability, to ensure reliable biomass estimations.

Supplemental Information

Supplemental Information 1 Dataset

Click here for additional data file.

We acknowledge the field and/or laboratory assistance of (in alphabetic order) T Amgren, E Anderberg, F Ek, P Jacobsson, G Johansson, C Jönander, L Näslund, O Pettersson, M van Regteren, S Skoglund, E Svartgren, V Thunell, C Åkerlund and M Åkerman. We thank the water right owners around each study bay for facilitating the field sampling. This study is a product of project Plant-Fish (www.plantfish.se).

Additional Information and Declarations

Competing Interests

Author Contributions

Data Availability

Göran Sundblad is an employee of AquaBiota Water Research. His work in relation to this study was fully funded by a grant from Formas, the Swedish research council for sustainable development.

Johan Eklöf conceived and designed the experiments, performed the experiments, analyzed the data, contributed reagents/materials/analysis tools, wrote the paper, prepared figures and/or tables.

Åsa Austin, Serena Donadi, Britas D.H.K. Eriksson and Göran Sundblad performed the experiments, contributed reagents/materials/analysis tools, reviewed drafts of the paper.

Ulf Bergström and Joakim Hansen performed the experiments, contributed reagents/materials/analysis tools, prepared figures and/or tables, reviewed drafts of the paper.

The following information was supplied regarding data availability:

The raw data has been supplied as Data S1.

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
