# Peer review of "Size matters: relationships between body size and body mass of common coastal, aquatic invertebrates in the Baltic Sea"

_PeerJ, doi:10.7717/peerj.2906_

## Round 0.1 · original submission · Minor Revisions

In addition to responding to the reviewers' comments below, would you also please attend to the following minor points:

line 138: change "taxa" to "taxon"
141, 143 & 144: de-italicize family names
193: change "that" to "which"
259: change to "First, while the rate of growth in length of mollusc shells typically decreases with age,..."
271: change to "which causes the small, osmotically stressed M. edulis to invest..."

Reviewer 1 ·

Basic reporting

formally ok, and in itself meeting al requirements, but see below.

specifics:
- We have a confusion of technical terms here:
Biomass = mass per unit of area or volume
Body mass = mass of one individual
The term "weight" is physically incorrect and should not be used
- Some relevant literature is neglected, e.g. Brey et al. 2010 Body composition in aquatic organisms – a global data bank of relationships between mass, element composition and energy content. Journal of Sea Research 64: 334-340.doi:10.1016/j.seares.2010.05.002

Experimental design

no experiments, sample treatment & statistics o.k.

Validity of the findings

My problem with this paper is that it looks at just one aspect of a multi-dimensional pattern. Even given that AFDM would be the most appropriate unit for mass - the plankton research community thinks differently, in strong favour of Corg - the approach neglects other factors that act on AFDM/DM besides size. These are, in the first instance, time (seasonal and year-to-year differences) and space (spatial differences, e.g. in alimentation level). Hence, AFDM-size-relationships vary distinctly in space and time (see literature).

If you want to make a case for estimating biomass through AFDM-size-relationships, you need to do two things: (i) to evaluate the “error-field” of this approach, and (ii) to compare your approach with others (general conversion factors). Regarding (i), the overarching question is accuracy of prediction, i.e. how does accuracy of prediction of the body mass of a particular individual relate to the width (in space and time) of the data base of the AFDM-size-relationship applied? Regarding (ii), the question is the effort required by both approaches to achieve comparable accuracy or the difference in effort required to achieve the same accuracy.
The way this paper does is, is, in my opinion, just not at the proper scientific level.

Additional comments

I presume that this paper is a spinoff from a larger study on macrobenthic communities that was not designed to address the question raised here, i.e. whether AFDM-size-relationships are a better and more efficient predictor of benthic biomass than general conversion factors. If such a general question is to be tested, a general approach has to be taken, decoupled from particularities such as the region the samples / data have been taken from.

·

Basic reporting

This paper is very well written – easy to follow, clear, unambiguous and has good use of professional English throughout. As far as I could tell it adheres to PeerJ standards and the structure is certainly a relevant structure in the discipline. I thought the resolution of the plots and taxonomic names associated with them in Figure 2 could be improved for clarity, but otherwise I was happy with the style, quality and use of figures and tables.

Experimental design

The research described is certainly original and well defined and the authors do a good job of explaining why it is relevant in the Introduction. I don’t have any concern regarding the rigor of the design or execution of the work undertaken and the study is repeatable (although I did feel there were a couple of references that could be added into the methodology to help on this – as covered in the General Comments section).

Validity of the findings

I think the findings are entirely valid and the discussion of these suitable and useful with some well stated conclusions drawn. I am happy with the data and statistics used.

Additional comments

This is a well written account of a well-designed and useful study that fills a gap that is quite rightly identified by the authors. I have a few specific comments that could be addressed to further improve the manuscript (see below) but overall I thought it was of a high standard.
Lines 61-63 – It would have been nice to cite some key references here as although I have no doubt the authors are correct, it would be useful for those less familiar to the field to be pointed in the right direction.
Lines 70-74 – Although I agree dry- or ash-free dry weight are the most appropriate measure to consider biomass in many circumstances, blotted wet weight has also been used in many analyses and we found that there are very good relationships between size and wet weight for many macrofaunal species (see Robinson et al., 2010 where significant relationships are reported for >200 species). Perhaps the authors might explain either why they think wet weight is not so useful, or at least refer to the work done in that area?
Line 88 – missing a semi-colon between the cited references
Lines 117-122 – Here the authors introduce the fact that the particular conditions of the Baltic Sea mean that size ranges and also size: weight relationships and AFDW/DW relationships could differ to conspecifics found in marine or freshwater. However, this issue is then not re-visited in the Discussion which seems like an important omission.
Lines 144-145 – The authors mention the size measurements are based on standard procedures, but give no source for these.
Line 302 – gap in references
Table 1 – did not fit on the page as printed
Figure 2 – poor quality resolution for species names above figures and in plots

---

## Round 0.2 · accepted · Accept

Thank you for submitting the revised MS, which I am pleased to accept. Could you please de-italicise the family names Chironomidae and Phryganeidae in Table 1 (this can be done at proofs stage).